# Expression of Oxidative Stress and Inflammation-Related Genes in Nasal Mucosa and Nasal Polyps from Patients with Chronic Rhinosinusitis

**DOI:** 10.3390/ijms23105521

**Published:** 2022-05-15

**Authors:** Hrvoje Mihalj, Josip Butković, Stana Tokić, Mario Štefanić, Tomislav Kizivat, Maro Bujak, Mirela Baus Lončar, Martina Mihalj

**Affiliations:** 1Clinical Department of Otorhinolaryngology, Head and Neck Surgery University Hospital Osijek, HR-31000 Osijek, Croatia; hrvoje.mihalj@gmail.com; 2Department of Otorhinolaryngology, Maxillofacial Surgery Faculty of Medicine University of Osijek, HR-31000 Osijek, Croatia; butkovic.josip@kbco.hr; 3Department of Oral and Maxillofacial Surgery, University Hospital Osijek, HR-31000 Osijek, Croatia; 4Department of Laboratory Medicine and Pharmacy, Faculty of Medicine, University of Osijek, HR-31000 Osijek, Croatia; stokic@mefos.hr; 5Department of Nuclear Medicine and Oncology, Faculty of Medicine, University of Osijek, HR-31000 Osijek, Croatia; mstefanic@mefos.hr (M.Š.); tomislavkizivat@gmail.com (T.K.); 6Clinical Institute of Nuclear Medicine and Radiation Protection, University Hospital Osijek, HR-31000 Osijek, Croatia; 7Department of Materials Chemistry, Ruđer Bošković Institute, HR-10000 Zagreb, Croatia; maro.bujak@irb.hr; 8Department of Molecular Medicine, Ruđer Bošković Institute, HR-10000 Zagreb, Croatia; 9Department of Dermatology and Venereology, University Hospital Osijek, HR-31000 Osijek, Croatia; 10Department of Physiology and Immunology, Faculty of Medicine University of Osijek, HR-31000 Osijek, Croatia

**Keywords:** chronic rhinosinusitis with nasal polyps, superoxide dismutase, peroxiredoxin-2, adenylate-cyclase-activating polypeptide receptor 1, bactericidal/permeability-increasing fold-containing family A, member 1

## Abstract

Chronic rhinosinusitis (CRS) is a prevalent, multifaceted inflammatory condition affecting the nasal cavity and the paranasal sinuses, frequently accompanied by formation of nasal polyps (CRSwNP). This apparently uniform clinical entity is preceded by heterogeneous changes in cellular and molecular patterns, suggesting the presence of multiple CRS endotypes and a diverse etiology. Alterations of the upper airway innate defense mechanisms, including antimicrobial and antioxidant capacity, have been implicated in CRSwNP etiology. The aim of this study was to investigate mRNA expression patterns of antioxidative enzymes, including superoxide dismutase (SOD) and peroxiredoxin-2 (PRDX2), and innate immune system defense players, namely the bactericidal/permeability-increasing fold-containing family A, member 1 (BPIFA1) and PACAP family members, particularly adenylate-cyclase-activating polypeptide receptor 1 (ADCYAP1) in nasal mucosa and nasal polyps from CRSwNP patients. Additional stratification based on age, sex, allergic comorbidity, and disease severity was applied. The results showed that ADCYAP1, BPIFA1, and PRDX2 transcripts are differentially expressed in nasal mucosa and scale with radiologically assessed disease severity in CRSwNP patients. Sinonasal transcriptome is not associated with age, sex, and smoking in CRSwNP. Surgical and postoperative corticosteroid (CS) therapy improves endoscopic appearance of the mucosa, but variably reverses target gene expression patterns in the nasal cavity of CRSwNP patients. Transcriptional cross-correlations analysis revealed an increased level of connectedness among differentially expressed genes under inflammatory conditions and restoration of basic network following CS treatment. Although results of the present study imply a possible engagement of ADCYAP1 and BPIFA1 as biomarkers for CRSwNP, a more profound study taking into account disease severity and CRSwNP endotypes prior to the treatment would provide additional information on their sensitivity.

## 1. Introduction

Chronic rhinosinusitis (CRS) is a prevalent, multifaceted inflammatory condition affecting the nasal cavity and the paranasal sinuses. Two fundamental clinical CRS manifestations are commonly differentiated according to the presence (CRSwNP) or absence (CRSsNP) of nasal polyps (NP), the edematous formations that often accompany chronic CRS symptoms like nasal congestion, posterior rhinorrhea, hyposmia, and facial pressure or pain. Several distinct immune cellular compartments and cytokine profiles have been more recently described within each CRS phenotypic subtype, suggesting the presence of multiple CRS endotypes [1,2]. Moreover, CRS immune responses were shown to vary across different geographic areas and ethnicities, further contributing to CRS heterogeneity [3]. In line with this, pronounced eosinophilia with type-2-based immune responses is consistently reported in Caucasian CRSwNP patients from Western countries [1,4,5,6], whereas non-eosinophilic CRSwNP with a type 1 and 3 immune profile is commonly demonstrated in their Asian counterparts [7,8,9,10]. High prevalence of comorbid allergies featuring increased local production of IgE against common aeroallergens, bacterial enterotoxins, or both [1] adds to the clinical complexity of various CRSwNP manifestations. Despite increased awareness of CRSwNP heterogeneity, associated molecular signatures underlying disease severity, tissue remodeling, or comorbid allergy remain, however, largely unknown.

Accumulating data support an important role of antimicrobial and antioxidant capacity of human nasopharyngeal and upper respiratory epithelium in CRSwNP etiology. In line with this, antioxidative enzymes, such as superoxide dismutase (SOD) and peroxiredoxin-2 (PRDX2), as well as bactericidal/permeability-increasing fold-containing family A, member 1 (BPIFA1) and PACAP family members, particularly adenylate cyclase activating polypeptide receptor 1 (ADCYAP1), have been identified as key accomplices in an effective, upper airway innate defense mechanisms. Accordingly, high BPIFA1 expression in normal, adult epithelium was shown to regulate airway surface liquid volume [11], neutrophil antibacterial activity [12], and antibiofilm function [13]. On the other hand, reduced BPIFA1 protein levels previously reported in mucosal epithelia of CRSwNP patients [14,15,16] may lead to insufficient maintenance of the epithelial barrier function and enhanced bacterial colonization. The inflammatory conditions in nasal mucosa of CRSwNP patients may, in addition, be regulated via ADCYAP1 signaling, generally associated with reduced mast cell degranulation and cytokine/chemokine production [17,18,19], downregulated expression of adhesion molecules [20], and diminished neutrophil infiltration [21]. Conversely, inflammation-induced oxygen free radical damage may foster nasal polyp formation and stimulate expression of antioxidative enzymes. The ADCAP1, BPIFA1, and SOD protein levels were previously found to be differentially expressed in nasal polyp and chronic sinusitis tissue of Chinese CRSwNP patients [22], but their role in the progression or treatment outcome of Caucasian CRswNP endotypes remains unaddressed.

We, thus, aimed to evaluate the transcriptional patterns of ADCAP1, BPIFA1, SOD, and PRDX2 genes across different locations of the nasal cavity, including nasal polyps, bulla ethmoidalis, and middle nasal turbinate tissue of CRSwNP patients. The potential contributing effects of altered transcriptional mechanisms on disease severity, comorbid allergic sensitivity, and postoperative corticosteroid treatment efficacy were examined as well.

## 2. Results

### 2.1. Patient Characteristics

CRSwNP patients were significantly older than controls (median (IQR): 31 (26–48) vs. 53 (51–62) years, controls *vs.* CRSwNP, Table 1) and more likely to be nonsmokers. The study population included more men than women, but the male/female ratio was similar in both cohorts. Both groups scored similarly on SNOT-20 test results, indicating no difference in health-related quality of life. The majority of CRSwNP patients had moderate to high preoperative Malm endoscopy and Lund–Mackay CT scores. IgE-mediated hypersensitivity was detected in six CRSwNP patients (24%). As expected, patients suffering from allergic rhinitis reported lower quality of life (25 (12–62) vs. 63 (49–83), *p* = 0.02, RIST neg. vs. pos., SNOT) and presented with more severe disease on endoscopy (5 (3–6) vs. 6 (5–6), *p* = 0.034, RIST neg. vs. pos., Malm score), but not on CT scans (10 (8–14) vs. 13 (11–18), *p* = 0.074, RIST neg. vs. pos., Lund–Mackay score). There was no correlation between SNOT-20 and CT scores. Age had no impact on Malm and CT scores (Spearman’s ρ = −0.06, *p* = 0.78, *n* = 25). Sinonasal cavities were colonized by commensal and pathogenic bacterial genera in most CRSwNP subjects (78%); in contrast, all nasal swabs remained sterile in the control group, probably reflecting the limits of the routinely applied microbial testing. The most frequently isolated bacterial species in the CRSwNP cohort were Klebsiella pneumoniae and Staphylococcus epidermidis, followed by sporadic identification of Staphylococcus aureus, Escherichia coli, Streptococcus hemolyticus group B, Morganella morganii, Enterobacter spp., Serratia marcescens, and Proteus mirabilis.

### 2.2. ADCYAP1, BPIFA1, and PRDX2 Transcripts Are Differentially Expressed in Nasal Mucosa and Scale with Radiologically Assessed Disease Severity and IgE-Positivity in CRSwNP Patients

We next focused on differential expression analysis, comparing the two study groups (CRS vs. healthy controls) at the RNA levels. To this purpose, we investigated gene expression profiles across the (para)nasal cavity by performing qRT-PCR on nasal polyps and/or mucosal samples from different locations (see Section 4.3). Generally, a double-coincidence discovery was used to extract those signatures that were shared by two or more sampling sites, thus allowing for better control of technically introduced sampling variations. We also explored the effect of patient-related differences, such as disease severity and IgE context, by dividing the CRS sample into two subsamples using IgE quantities and Lund–Mackay CT scores.

This revealed a remarkable complexity of gene expression that occurs on several levels.

Despite modest ICC (ICC(2,k) = 0.43–0.47, ICC(2,k) $\begin{array}{c} PRDX2\\ max \end{array}$ = 0.72), the overall expression of target genes was largely consistent across the regions (Figure 1), with most variability arising from differences in CRSwNP severity on Lund–Mackay CT scans (Figure 2). In case–controls comparisons, ADCYAP1 was uniformly and significantly the most downregulated gene at all sampled sites in CRSwNP patients, reaching a nadir in severely affected individuals. Conversely, BPIFA1 transcripts were consistently expanded in all sinonasal regions, peaking in MNT mucosa of CRSwNP patients when compared to LNT tissue of healthy controls (Figure 1). Single-cell RNA sequencing (scRNA-seq) data from human nasal and lung tissue showed almost selective expression of BPIFA1 and ADCYAP1 in goblet/glandular epithelial cells and mast cells, respectively [23,24,25,26], Figure 3), thus providing an insight into the cell-type-specific gene expression patterns.

PRDX2 and SOD1 levels declined steadily with increasing Lund–Mackay CT scores (Appendix A); as a result, significantly lower expression of PRDX2 and SOD1 was observed in BE and MNT mucosa of CRSwNP patients having high Lund–Mackay CT scores (>12, all nonsmokers), but not in mildly affected patients having low CT scores (≤12) (Figure 2). We also checked the sensitivity of the results against a grid of CT cut-off scores (≥10, ≥12), confirming their robustness (data not shown). In contrast to ADCYAP1 and BPIFA1, expression of PRDX2 was relatively ubiquitous in human nasal and lung tissue, whereas SOD1 expression was the highest in ciliated epithelial cells, lymphatic endothelial cells and mesenchymal cell types, including CRTHC1+ fibroblasts/smooth muscle cells (Figure 3). Notably, the loss of mature ciliated cells is a common event in CRSwNP and Th2 inflamed epithelia [27,28,29]; moreover, the surviving metaplastic ciliated cells are known to downregulate many genes involved in xenobiotic metabolism and oxidoreductase activity [30], including SOD1 [29].

The presence or absence of IgE-mediated hypersensitivity was also linked to variability in mucosal ADCYAP1 and PRDX2 expression. A significant downregulation was consistently evident across all sampling sites in RIST-positive but not RIST-negative CRSwNP patients when compared to control tissue from RIST-negative controls (Figure 4), a finding that could not be explained by radiologically more severe disease in RIST-positive patients. At this point, the lack of RIST-positive control individuals precluded a more detailed investigation into this issue.

Overall, BE and MNT mucosa were similar in terms of ADCYAP1, PRDX2, and SOD1 down-expression in radiologically severe CRSwNP, whereas polypous tissue showed a more restricted pattern of transcriptional alterations (ADCYAP1, PRDX2), predominantly in RIST-positive CRSwNP. Altogether, there was evidence of potentially regionalized gene expression programs in (sino)nasal mucosa undergoing polypoid maturation, a part of this heterogeneity emerging in relation to the IgE-primed environment. We note, however, that this IgE-(+) bin should be considered with caution.

### 2.3. Sinonasal Transcriptome Is Co-Modified by Smoking in CRSwNP

In addition to disease severity and atopy, contributions from age, sex, microbiome, and smoking may be expected. In this section, we thus consider the effect of these covariates. No significant relationship was observed between target gene expression, sex, and nasal microbiome. A modest correlation between age and ADCYAP1/BPIFA1 expression emerged in a pooled analysis of patients with CRSwNP and healthy individuals (Appendix A), but no compelling evidence of age-related trends was observed within either group when tested separately (data not shown). In addition, age-related accumulation of SOD1 and PRDX2 transcripts was observed in smokers, although it did not amount to significant difference between the smokers and nonsmokers, Appendix A. Smokers also largely belonged to healthy controls, which helped reduce the spillover effect between smoking and CRSwNP. To correct for age-related confounding, a general linear model adjusting for age was fitted at individual-level data, showing significant residual associations of radiologically severe CRSwNP with ADCYAP1, PRDX2, and SOD1, but not BPIFA1 levels in nonpolypous mucosa (BE and MNT) (Appendix A). For polypous tissue, the residual association with disease severity was limited to the ADCYAP1 gene. Secondly, a robust residual association with RIST-positive CRSwNP persisted for ADCYAP1 and PRDX2 levels in replicate samples coming from polypous tissue and MNT mucosa. The tightness of the Lund–Mackay CT score–ΔΔCt sequence (Appendix A) is a testimony to this model; had there been any excess age-related variations, this would have increased the resulting scatter beyond what is seen. Overall, we conclude that target gene expression was relatively robust to age differences; however, we note that a direct comparison of age-matched control individuals with the CRSwNP cohort, or even a prospective study, would be better suited to definitively answer this question.

### 2.4. Surgical and Postoperative Corticosteroid (CS) Therapy Improves Endoscopic Appearance of the Mucosa, but Variably Reverses Target Gene Expression Patterns in Nasal Cavity of CRSwNP Patients

We next sought to understand the temporal evolution of target gene expression in response to specific medical treatment. Paired MNT samples were collected prior to and six months after the surgical and postoperative corticosteroid treatment to examine dynamics of sinonasal gene expression in CRSwNP patients. In total, 16 pairs of MNT mucosa were available for this analysis, mostly from IgE-negative patients (12/16) having low pretreatment Lund–Mackay CT scores (5/16). Overall, we note a highly variable transcriptional response under treatment, despite the normal appearing mucosa and evidence of complete remission in all subjects on endoscopic reassessment. ADCAYP1 and BPIFA1 expression patterns were both reversed under treatment, reaching control levels (Figure 5A). For SOD1, post-treatment expression levels did rise in normal-looking mucosa (from 1.26 (0.47–2.09) to 1.6 (1.15–2.22), fold change, *p* = 0.058, Wilcoxon’s test), surpassing healthy control levels (Figure 5B). However, factors predicting transcriptional variations in response to CS treatment could not be identified in this study. A single sampling time point contributes to this uncertainty, obscuring the differences between early and long-term steroid-induced changes in nasal mucosa.

### 2.5. Transcriptional Cross-Correlations Vary across Nasal Tissues and with Regard to Case–Control Status in CRSwNP

Innate immunity and redox defenses are deeply and dynamically intertwined in stressed mucosa [31,32,33], and CRSwNP is probably no exception to this [34]. The details, however, remain elusive, as past research on many of these facets is limited. In order to gain insight into the gene co-expression patterns and their changes with regard to sinonasal spatial niche and case–control status, an intergenic correlation analysis was performed (Figure 6). A consistent pattern of strong PRDX2-SOD1 co-expression was observed across all sampling sites in both cases and controls, particularly in LNT, CS-treated MNT, and BE CRSwNP samples. In addition to this basic pattern, a moderate to strong, positive and negative rewiring of the PRDX2|SOD1 module with the ADCYAP1 and BPIFA1 gene expression, respectively, was observed in untreated MNT and polypous CRSwNP mucosa. From this, we conclude that sinonasal transcriptional remodeling is accompanied by an increased level of connectedness among differentially expressed genes under inflammatory conditions. Secondly, CS treatment was apparently associated with restoration of the basic network pattern, as indicated by correlative evidence encompassing a small set of genes from CS-treated and untreated MNT mucosa.

## 3. Discussion

In this study, we evaluated transcriptional patterns of several genes involved in antioxidative defense and innate immune system across different locations within the nasal cavity of CRSwNP patients. Here we demonstrate differential expression of ADCYAP1, BPIFA1, SOD, and PRDX2 transcripts in nasal mucosa, associated with radiologically assessed disease severity and comorbid allergic hypersensitivity. Sinonasal transcriptome was not associated with age, sex, and smoking in CRSwNP. Surgical and postoperative corticosteroid (CS) therapy improved endoscopic appearance of the mucosa, but variably reversed target gene expression patterns in the nasal cavity of CRSwNP patients. Transcriptional cross-correlation analysis revealed an increased level of connectedness among differentially expressed genes under inflammatory conditions and restoration of basic network following CS treatment.

Prior to therapy, BPIFA1 expression levels were elevated across all sinonasal regions, particularly in MNT tissue and nasal polyps of severely affected, RIST-negative CRSwNP patients. Upregulated BPIFA1 expression is known to be sustained under inflammatory conditions, as exemplified in the epithelium of the proximal and distal cystic fibrosis airways [35], and in in vitro airway cultures of Mycoplasma pneumoniae [36] and Klebsiella pneumoniae [37]. In this regard, BPIFA1 was shown to inhibit growth of Mycoplasma pneumonia [36], act directly against Gram-negative bacterial cell wall integrity and biofilm formation [38], as well as a chemoattractant that recruits macrophages and neutrophils to the site of infection [13], thereby enhancing epithelium-mediated innate host defenses. Conversely, in an allergic, IL-13-enriched, Th2 inflammatory milieu, BPIFA1 levels were found to be significantly diminished [36,39], which may prove detrimental to the host immunity response. Reduced BPIFA1 protein levels have been, indeed, previously described in CRSwNP patients [14,15,40], and associated with increased bacterial colonization and persistent or recurrent sinusitis. Polyp recurrence and increased rate of revision surgery are commonly featured in more severe CRSwNP endotypes, characterized by type 2 inflammation, comorbid allergy, local eosinophil infiltration, and high IgE tissue levels [41,42,43]. In our sample cohort, comorbid allergic/IgE-mediated hypersensitivity was detected in only six patients, whereas the remaining majority manifested IgE-negative disease form, plausibly underlining increased BPIFA1 transcriptional signature observed in our study subjects. Differences in eosinophil/neutrophil count, age, and preoperative CS therapy of CRSwNP examinees could be an additional source of confounding between our and previous studies and warrant caution in the data analysis and interpretation.

Other aspects of epithelial host defense were also impaired in nasal mucosa of our CRSwNP cohort. Steady decline in ADCAYP1 expression was evident across all sampled nonpolypus (MNT, BE) and polypus areas, and strongly associated with disease severity and comorbid IgE-mediated hypersensitivity. Lack of ADCYAP expression has been previously implicated in murine endotoxin-induced airway inflammation, airway hyperreactivity, histopathological changes, neutrophil, and macrophage lung accumulation [44]. In human nasal cavity, ADCYAP was shown to augment airway resistance by inhibiting histamine-induced neutrophil recruitment, plasma leakage, and IL-1RA expression levels in nasal lavage [21], indicating the protective, anti-inflammatory role of ADCYAP expression in nasal mucosa. It is, thus, not surprising that we and others noticed diminished ADCYAP expression in nasal tissue of CRSwNP patients [22,40]. Of interest, recent RNASeq study confirmed significantly reduced ADCYAPR1 mRNA levels in polypus tissue of CRSwNP patients with and without comorbid asthma [45]. Our results support previous reports on ADCYAP decline in CRSwNP and highlight the significant potential of ADCAYP treatment in inflammatory and allergic conditions of both polypus and nonpolypus tissue in CRS patients. Moreover, the normalization of ADCYAP expression observed here post CS treatment supports its therapeutic significance and complements current understanding of postoperative long-term CS treatment outcomes in CRSwNP.

Next to the disturbed expression of epithelium-derived inflammatory mediators, we found evidence of severely diminished antioxidative capacity in CRSwNP, which was only partially remediated by postoperative CS therapy. Expression of SOD1 was most significantly reduced in MNT and BE, but not polypus tissue of severely affected CRSwNP patients, while diminished PRDX2 mRNA levels persisted across all sampled sites, dominantly in RIST-positive CRSwNP examinees. Similar findings in terms of reduced SOD expression in nonpolypus nasal tissue have been previously reported in CRSsNP patients [46]. In contrast, increased SOD1 levels have been described in nasal polyps of nonallergic CRSwNP subjects, reminiscent of SOD1 differences observed in nasal polyps of our RIST-positive and RIST-negative examinees. Interestingly, a higher percentage of SOD1-positive epithelial cells was previously demonstrated in non-eosinophilic compared to eosinophilic CRSwNP endotype [47], suggesting an active role of eosinophil-induced inflammatory environment in the deterioration of the nasal oxidation-reduction system. Comorbid IgE hypersensitivity similarly affected PRDX2 expression across all sampled sites, supporting the notion of compromised antioxidative potential in severe, IgE-primed disease settings. In line with this, several previous studies reported increased oxidative stress and reduced SOD1 capacity in CRSwNP patients [48,49]. Reduced PRDX2 protein expression has been also established in asthma patients [50]. Improvement of CRS clinical symptoms following application of steroid therapy supplemented with antioxidants confirms significance of valid antioxidative protection in CRSwNP treatment [49].

For BPIFA1, ADCYAP1, and PRDX2 genes, surgical and long-term local CS treatment reversed target gene expression. Some discrepancies remain, however, as CS treatment was less successful in establishing normal SOD1 homeostasis in otherwise healthy-looking mucosa, supporting the need for additional, longitudinal assessments of CS treatment outcomes in CRSwNP. Our study would also benefit from a higher number of examinees, particularly healthy controls of older age and RIST-positive status, to establish the true effect size of aging and IgE-primed response in shaping nasal mucosa transcriptome. For that matter, further characterization of immune cell variety, phenotype, and number, complemented with immunohistochemical validation of protein candidates encoded by mRNA studied here, would be required in the next step to elucidate the most promising therapeutic targets.

Overall, we bring forth evidence of regionalized transcriptional programs, differentially altered in mild and severely affected CRSwNP patients, supporting a role for gradual decline in innate host defenses and antioxidant capacity in disease evolution. Observed transcriptional alterations were partly recovered under CS therapy, which proved efficient in restoration of epithelium-derived anti-inflammatory mediators, but less successful in rescuing sinonasal antioxidative potential. Future multidimensional studies in clinically well-characterized CRSwNP cohorts are required to complement our findings and identify pathologic mechanisms underlining altered transcriptional patterns and steroid-induced changes in nasal mucosa.

## 4. Materials and Methods

### 4.1. Design

This study explored patients with CRSwNP that were treated with functional endoscopic sinus surgery (FESS). Patients were treated during a 12-month period, with conclusion on September 2013, at the Department of Otorhinolaryngology and Head and Neck Surgery (DOHNS) of the University Hospital Osijek (Croatia). As a control group, patients that had nasal septoplasty for correction of anatomical variations, such as hypertrophic lower turbinate or deviated nasal septum, were used. This prospective study acquired approval from the Ethics committee of the Faculty of Medicine University of Osijek (Croatia) and the Ethics committee of the University Hospital Osijek (No. 25-1:11709-2/2010 issued on 13 December 2010); it had been planned and performed according to the principles of the Declaration of Helsinki. All patients have signed the informed consent.

### 4.2. Subjects

A total of 25 patients with CRSwNP and 24 patients in the control group were treated at the DOHNS and selected to participate in this study. Patients were firstly examined by rhinoscopy and endoscopy, and followed up by standard laboratory diagnostic, blood tests, total and specific IgE, nasal smear for microbiological assessment, and eosinophils. All patients were treated in accordance with European Position Paper on Rhinosinusitis and Nasal Polyps [51]. Before surgical treatment of CRSwNP, patients were treated with conservative therapy, in which patients were instructed to use intranasal steroid therapy, and antibiotics were used as needed. Use of local (mometasone furoate) and systemic steroids (15, 10, and 5 mg of oral prednisone) for three weeks prior to surgical treatment was consistent in all patients. If the results of conservative treatment were not satisfactory, patients were treated with FESS and instructed to continue the use of local intranasal steroids for no less than six months. Indication for FESS in patients with CRSwNP was a deficient response to conservative treatment during a six-month period, with continuation of symptoms and reduction in quality of life. In the control group, insufficient nasal airflow was the indication for septoplasty. Samples of middle nasal turbinate (MNT-0), Bulla ethmoidalis (BE), and nasal polyps (NP) were collected during operation in patients with CRSwNP, a control sample of middle nasal turbinate was taken six months after operation. In the control group, a sample from lower turbinate was taken during operation.

Wigand technique was used for FESS operation on patients with CRSwNP, and Cottle technique was used for septoplasty on control patients, both with Karl Storz and Wolf operating equipment. Samples for the quantitative PCR analysis were obtained by the same surgeon during operation from the corresponding regions using standard Blakesley forceps (size 3, cup 5 × 15 mm, Karl Storz, Tuttlingen, Germany). Control samples from patients with CRSwNP were taken 6 months after surgery with local anesthesia. Exclusion criteria for all patients were: under 18 years of age, primary ciliary dyskinesia, systemic diseases involving the nose (sarcoidosis, cystic fibrosis, Kartageners syndrome, and Wegener’s granulomatosis), pregnancy, and active oncological treatment. Additional exclusion criteria were set for the control group and included proven allergy, or bacterial or fungal colonialization of the nasal cavity. These exclusion criteria were set for the purpose of assessing allergy and bacterial colonization influence on target genes in patients with CRSwNP. General information about patients was also collected: age, gender, history of smoking and asthma, CT Lund–Mackey and Malm classification–total nasal endoscopy score (both only for CRSwNP patients) SNOT 20 analysis, and the follow-up of the surgical procedure and the postoperative period.

### 4.3. Quantitative Real-Time Polymerase Chain Reaction

Samples were collected from corresponding localizations during the surgery at the DOHNS. Tissue samples were put in RNAlater solution (Qiagen, Hilden, Germany) for up to 4 h at room temperature, at +4 °C overnight, and stored at −80 °C until RNA isolation. Total RNA was isolated according to manufacturer’s protocol with RNeasy Mini Kit (Qiagen, Hilden, Germany). Using High-Capacity cDNA Reverse Transcription Kit (Applied Biosystems, Waltham, Massachusetts, USA), 1.5 µg of RNA was, according to manufacturer’s protocol, transcribed into cDNA. Quantitative real-time PCR was performed using specific primers and SYBR Green I (Invitrogen, USA) on 7300 Real-Time PCR System (Applied Biosystems, SAD). Target genes were ADCYAP1, BPIFA1, SOD1, and PRDX2, and the housekeeping genes were: HPRT, 18S, GUS, YWHAZ, and RPL13A. A list of all genes with sequences is shown in Table 2. Expression levels of investigated transcripts were normalized relative to the 18S reference gene, selected according to the lowest stability measure (M = 1.170) determined via NormFinder algorithm. qRT-PCR cycling conditions were: 95 °C for 3 min polymerase activation, 40 cycles of 95 °C for 1 min, the specific annealing temperature specific for primer pairs according to Table 2 for 30 s, and 72 °C for 30 s of elongation. Melting curve analysis was performed to confirm single product amplification. Gene expression was analyzed using REST © software (ΔΔCt method) and normalized to stable housekeeping genes. Changes were represented as log2 (fold change).

### 4.4. Clinical Assessment of Disease Severity and Life Quality

#### 4.4.1. CT Lund–Mackey Score

The Lund–Mackay score is a widely used method for radiologic staging of chronic rhinosinusitis. Preoperative CT scans were analyzed for mucosal abnormalities of the maxillary, anterior ethmoid, posterior ethmoid, frontal and sphenoid sinuses, and, additionally, for the occlusion of ostiomeatal complex (OMC). The Lund–Mackay staging system assigns a value of 0 (no abnormality), 1 (partial opacification), or 2 (total opacification) for each sinus group, while the ostiomeatal complexes were scored bilaterally as 0 (not occluded) or 2 (occluded). The maximal CT grading score was 24.

#### 4.4.2. Malm Classification

Staging of nasal polyposis was conducted based on endoscopic findings according to the Malm score: grade 0—no polyps; grade 1—polyps in the middle meatus, not reaching below the inferior border of the middle turbinate; grade 2—polyps reaching below the inferior border of the middle turbinate but not the inferior border of the inferior turbinate; grade 3—large polyps reaching to or below the inferior border of the inferior turbinate or polyps medial to the middle turbinate. The total endoscopy score was calculated as a sum of scores given for each nasal cavity, ranging from 0 to 6.

#### 4.4.3. SNOT 20 Questionnaire

SNOT-20 questionnaire was used to assess various aspects of life quality, including disease-related symptoms (i.e., runny nose, postnasal discharge, etc.), functional limitations (i.e., sleeping difficulties), and emotional consequences. For each of 20 questions, patients were asked to give a score ranging from 0 (no problem) to 5 (severe problem). The total score is calculated as the sum of all given scores ranging from 0 to 100 and reflects life quality.

### 4.5. Nasal Smear

To acquire nasal smear, both nostrils were swabbed, smeared on a glass slide, and air dried. Glass slides were processed at the Laboratory for Cytology at the Clinical Department for Cytology of the University Hospital Osijek (Croatia) by standard eosinophilic stain and examined under light microscope.

### 4.6. Nasal and Sinus Swabs

Nasal and sinus swabs were performed for bacterial cultivation and identification. For the patients with CRSwNP, sinus swabs were taken during surgery, and, for the control group, nasal swabs were taken before surgery. Flocked swabs (Copan Italia S.p.A., Brescia, Italy) were used to maximize bacterial yield. All swabs that came in contact with nasal vestibule while the samples were taken were excluded to avoid contamination. There was no purulent secretion in the ethmoid sinus among the patients participating in the study. Samples were routinely analyzed in the laboratory for microbiology at the Institute of Public Health Osijek-baranja County (Croatia).

### 4.7. Statistical Methods

Continuous data are presented as median with interquartile range. Patients were arbitrarily binarized as having mild (≤12, *n* = 16) or severe disease (>12, *n* = 9) by median Lund–Mackay CT score. We used intraclass correlations coefficient (ICC) to estimate intra-individual transcriptional similarities across the nasal cavity (psych library, two-way random effect, average_random_raters, ICC(2,k), McGraw and Wong convention) [52]. In general, a modest correlation was noted across different locations within the same individual, prompting us to carefully proceed with the nonparametric ANOVA (Kruskal–Wallis test, false-discovery-rate-corrected Conover’s *post hoc* test; multcomp, PMCMRPlus package), and general linear model (Tukey’s contrasts, Westfall adjustment; lme4, rstatix package). For the latter, log2-transformed values were used to approach normality and homogeneity of residuals. For two independent groups, the Mann–Whitney test was used. Wilcoxon’s signed-rank test for difference in medians was applied to paired samples, whereas Fisher’s exact test was applied to contingency tables. Pairwise correlations were assessed by Spearman’s rank test. Two-tailed *p*-value < 0.05 was considered significant. All analyses were performed in R software v4.0.2 (www.r-project.org) using RStudio v 1.2.5001 environment (Rstudio Inc., Boston, MA, USA).

## Figures and Tables

**Figure 1 ijms-23-05521-f001:**
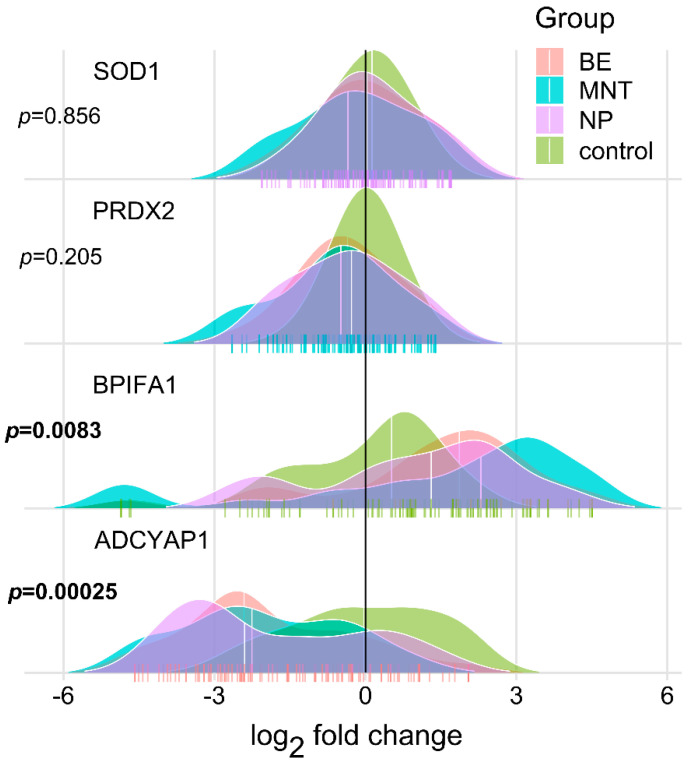
Ridgeline plots showing gene expression values (fold change, log2-scale) of indicated genes across the sampling sites. For differentially expressed transcripts (*p* < 0.05, two-tailed Kruskal–Wallis test), global *p*-values are depicted in bold. Each “|” point shape represents an individual.

**Figure 2 ijms-23-05521-f002:**
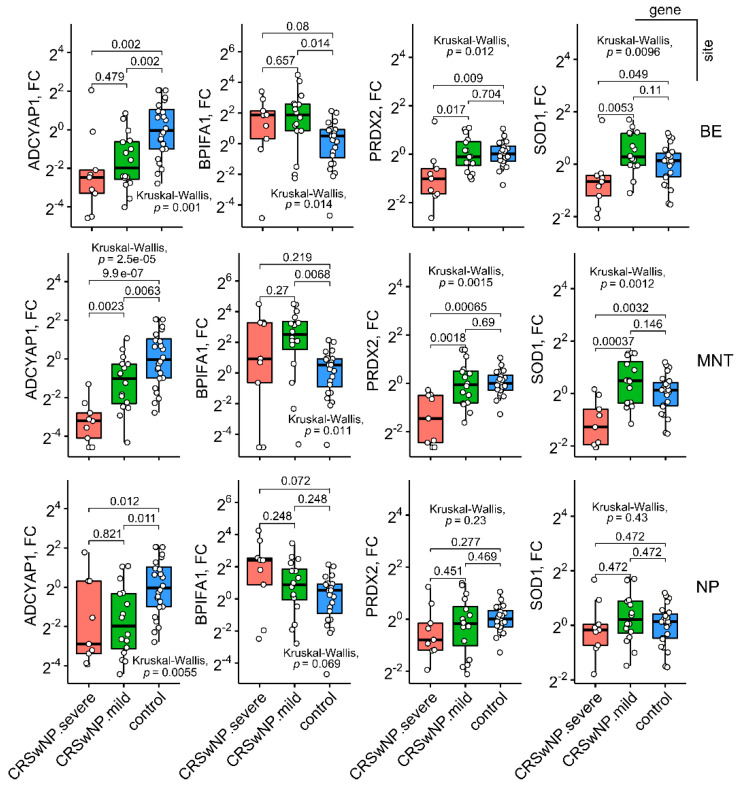
Boxplots showing gene expression (fold change (FC), log2-scale) in annotated groups, according to the Lund–Mackay CT score (two-tailed Kruskal–Wallis test, Conover’s post hoc test, false discovery rate correction). Horizontal lines represent median with interquartile range. *p* < 0.05 was considered significant. BE: bulla ethmoidalis, MNT: middle nasal turbinate, NP: nasal polyp. *n* (CRSwNP.severe) = 9, *n* (CRSwNP.mild) = 16, *n* (control) = 24.

**Figure 3 ijms-23-05521-f003:**
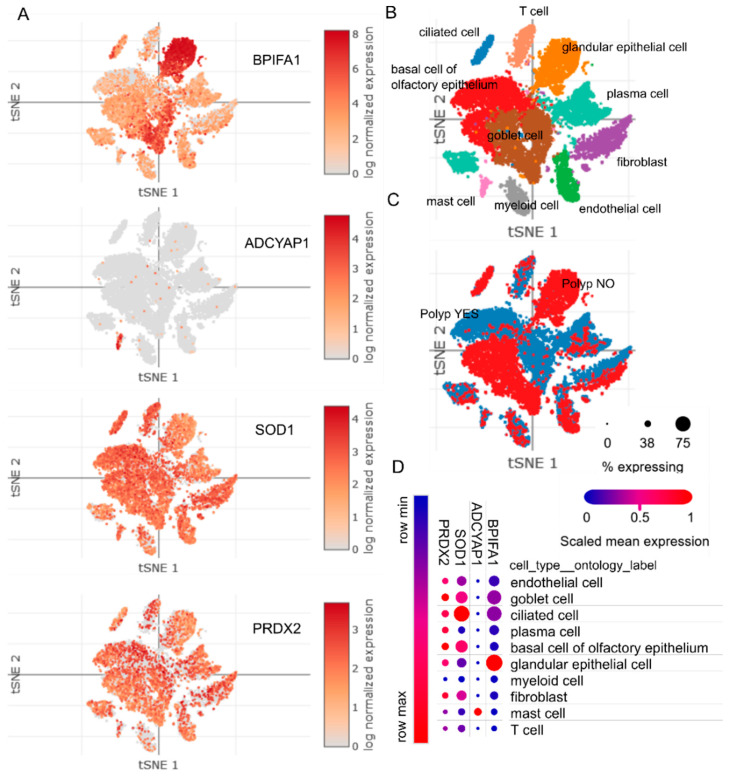
Single-cell landscape of human surgical chronic rhinosinuitis samples. (**A**) t-distributed Stochastic Neighbor Embedding (tSNE) of jointly analyzed single-cell transcriptomes of 18,036 cells from 12 chronic rhinitis samples annotated by (**B**) cell type and (**C**) disease status. (**D**) Dot plot of selected transcripts mapped onto cell types across all samples. Image credit: The Broad Institute (Cambridge, MA, USA) Single Cell Portal ID SCP235 (singlecell.broadinstitute.org/single_cell accessed on 14 April 2022) [27].

**Figure 4 ijms-23-05521-f004:**
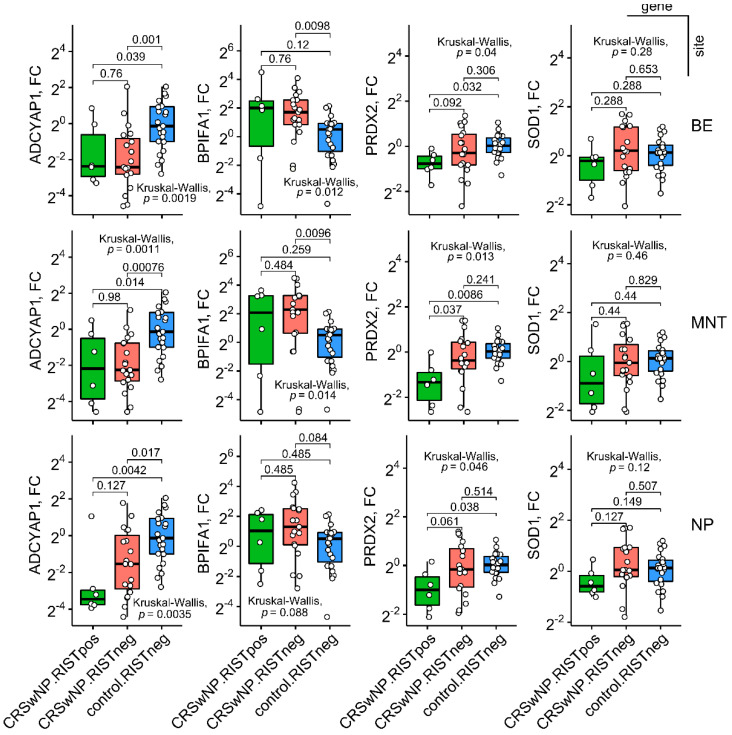
Boxplots showing gene expression (fold change (FC), log2-scale) in annotated groups, according to serum radioimmuno assay (RIST) of immunoglobulin E positivity (two-tailed Kruskal–Wallis test, post hoc Conover’s test, false discovery rate correction). Horizontal lines represent median with interquartile range. *p* < 0.05 was considered significant. BE: bulla ethmoidalis, MNT: middle nasal turbinate, NP: nasal polyp. *n* (CRSwNP.RIST.pos) = 6, *n* (CRSwNP.RIST.neg) = 19, *n* (control.RIST.neg) = 23.

**Figure 5 ijms-23-05521-f005:**
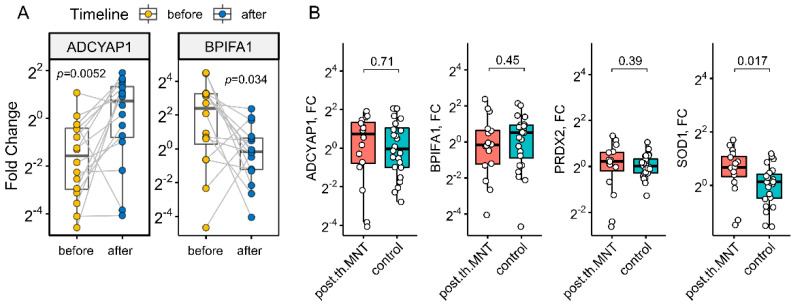
Boxplots showing gene expression (fold change (FC), log2-scale) in annotated groups (**A**) before and after treatment (*n* = 16, two-tailed paired Wilcoxon test, middle nasal turbinate) (**B**) in treated (middle nasal turbinate mucosa) and control samples (two-tailed Mann–Whitney test). Horizontal lines represent median with interquartile range. *p* < 0.05 was considered significant.

**Figure 6 ijms-23-05521-f006:**
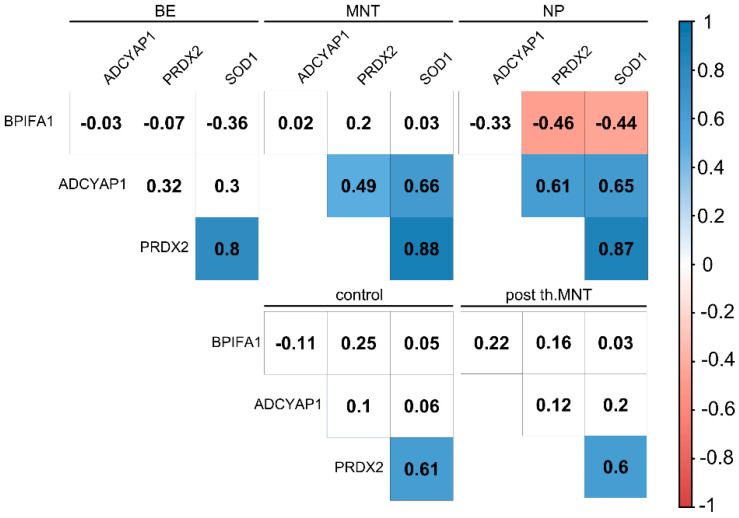
Correlation heatmaps depicting bulk gene expression in annotated groups (Spearman’s correlation coefficient). Significantly correlated genes are depicted by colored rectangles, whereas a blank cell denotes insignificant *p*-value as per level of 0.05. Sidebar depicts the correlation coefficient. BE: bulla ethmoidalis, MNT: middle nasal turbinate, NP: nasal polyp, post.th: post-treatment.

**Table 1 ijms-23-05521-t001:** Patients’ characteristics.

Characteristics	CRSwNP (*n* = 25)	Control (*n* = 24)	*p*
Age (yrs, min-max) *	53 (51–62)	31 (26–48)	1 × 10^−6^
Sex (m/f) **	15/10	13/11	0.776
Smoking (yes/no) **	3/25	11/23	0.012
SNOT-20 *	32 (15–64)	34 (19–52)	0.936
Malm score	5 (3–6)	NA	-
Lund–Mackay CT score	12 (9–15)	NA	-
Allergic disease (yes/no) **	6/19	1/23	0.074

Data are presented as proportions or median-IQR and were compared using * Mann–Whitney test or ** Fisher exact test, respectively. *p* < 0.05 was considered statistically significant. NA: not applicable, SNOT-20: Sino-Nasal Outcome Test-20, CT: computed tomography.

**Table 2 ijms-23-05521-t002:** Oligonucleotide primers used for qRT-PCR analysis.

Gene		Sequence 5′-3′	Optimized PCR Condition (Annealing Temp/MgCl_2_)
**ADCYAP1**	For	GAATTGGATTTGCATTCCCAGGCG	64 °C/3.5 mM
	Rev	AGGCATAGACCGAATGCCTCTGTT	
**PRDX2**	For	CCTTCCSGTACACAGACGAGCA	60 °C/3.0 mM
	Rev	CTCACTATCCGTTAGCCAGCCT	
**SOD1**	For	CTCACTCTCAGGAGACCATTGC	60 °C/3.0 mM
	Rev	CCACAAGCCAAACGACTTCCAG	
**PLUNC**	For	GGTTCTCAGAGGCTTGGACATC	65 °C/3.5 mM
	Rev	CCTTCCTGGAAGGCTTAGACCT	
**18S rRNA**	For	GTAACCCGTTGAACCCCATT	59 °C/3 mM
	Rev	CCATCCAATCGGTAGTAGCG	

**ADCYAP1**: adenylate cyclase-activating polypeptide 1 gene, same as PACAP—pituitary adenylate cyclase-activating polypeptide gene; **PRDX2**: peroxiredoxine 2 gene, same as NKEF-B—natural killer cell enhancing factor B gene; **SOD1**: superoxide dismutase 1 gene; **PLUNC**: palate, lung, and nasal epithelium clone protein gene, same as **BPIFA1**: BPI fold-containing family A member 1; **18S**:18S ribosomal RNA.

## Data Availability

Data available on request from the authors.

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
