# Peer review of "Expression of Oxidative Stress and Inflammation-Related Genes in Nasal Mucosa and Nasal Polyps from Patients with Chronic Rhinosinusitis"

_ijms, 2022, doi:10.3390/ijms23105521_

Round 1

Reviewer 1 Report

【General comments】

This manuscript is an article about analysis of expression of oxidative stress and inflammation related genes in nasal mucosa and nasal polyps in CRSwNP.

This is a very well-described review. However, there are some improvements that should be made. 

【Specific comments】

(1)figure4

Authors showe gene expression level according to RIST positivity in Figure4.

CRSwNP is also closely associated with eosinophil inflammation.

Please show gene expression according to level of blood eosinophils.

Please let me have a comment.

(2) Page9, Line393-396

Authors say that BPIFA1 was shown to act directly against bacterial cell wall integrity and airway epithelia biofilm formation. And in LINE393, up-regulated BPIFA1 expression is known to be sustained under inflammatory conditions, as exemplified in the epithelium of the proximal and distal cystic fibrosis airways, and in in vitro airway cultures of Mycoplasma pneumoniae.

But I think that Mycoplsma pneumoniae don’t have bacterial cell wall.

Please let me have a comment.

I hope that my comments are useful for the improvement of this manuscript.

Author Response

Dear Editor,

We thank you for the opportunity to submit a revised version of the manuscript titled “Expression of Oxidative Stress and Inflammation Related genes in Nasal Mucosa and Nasal Polyps from Patients with Chronic Rhinosinusitis” (ijms-1707010 ) to the International Journal of Molecular Sciences. We are grateful to the reviewers for their comments and valuable suggestions regarding our manuscript. Here, we have enclosed a revised version of the manuscript in response to the reviewer comments, which have been very helpful in improving the manuscript. Included below is a point-by-point description of our responses to reviewer’s comments. Changes made in the manuscript are highlighted in red. We cordially hope that you and the reviewers find this revised manuscript acceptable for publication in your journal.

Martina Mihalj, MD, PhD

Reviewer 1

General comment

This manuscript is an article about analysis of expression of oxidative stress and inflammation related genes in nasal mucosa and nasal polyps in CRSwNP.

This is a very well-described review. However, there are some improvements that should be made. 

Specific comments

  1. Figure4 Authors show gene expression level according to RIST positivity in Figure4. CRSwNP is also closely associated with eosinophil inflammation. Please show gene expression according to level of blood eosinophils. Please let me have a comment.

We thank the reviewer for his comment. Unfortunately, peripheral blood eosinophil counts were not included in our study design; hence the data are not available to the authors. However, based on the data available from the literature, total IgE levels correlate positively with peripheral blood eosinophil counts in several allergic diseases, including asthma, atopic dermatitis and allergic rhinitis (Altaii HA et al., 2022). In addition, in our study we found 100% congruence between RIST (total IgE), RAST (specific IgE) and nasal smear for eosinophils positivity. Based on these facts, we consider that the stratification of CRSwNP patients according to RIST test allowed us to distinguish between allergic and non-allergic patients most likely corresponding to eosinophilic and non-eosinophilic CRSwNP, respectively.

  1. Page9, Line393-396 Authors say that BPIFA1 was shown to act directly against bacterial cell wall integrity and airway epithelia biofilm formation. And in LINE393, up-regulated BPIFA1 expression is known to be sustained under inflammatory conditions, as exemplified in the epithelium of the proximal and distal cystic fibrosis airways, and in in vitro airway cultures of Mycoplasma pneumoniae. But I think that Mycoplsma pneumoniae don’t have bacterial cell wall. Please let me have a comment.

We acknowledge Reviewer’s remark and regret unprecise phrasing. Our initial intent was to point out variety of BPIFA1 antimicrobial mechanisms in general model of bacterial infection, referring not only to Mycoplasma pneumoniae, whose growth and function are inhibited by BPIFA1 (Chu et al., 2007), but to gram-negative bacteria as well (such as Klebsiella pneumoniae), which are negatively affected by BPIFA1-dependent changes in bacterial cell wall permeability (H. Zhou et al., 2006) and surface tension of airway fluids (Liu et al., 2013). To express our thoughts more clearly, we suggest the following amendments in the revised manuscript:

In this regard, BPIFA1 was shown to inhibit growth of Mycoplasma pneumoniae [25], act directly against gram-negative bacterial cell wall integrity [27] and biofilm formation [26], as well as a chemoattractant that recruits macrophages and neutrophils to the site of infection [13], thereby enhancing epithelium-mediated innate host defenses. (line 417-421)

Reviewer 2 Report

I thank the editor for invitation to review this manuscript.

The study is very interesting, however there are some observations to be made.

The results and discussion are presented in a confusing way.

The authors take into consideration numerous parameters of evaluation and continue with numerous and varied comparisons.

All this makes the discourse redundant and difficult to follow.

Authors should discuss the significance of the data in more detail.

Other considerations:

Section 4.2 Subjects: number of patients shown in the table does not correspond to that in the text.

Was a cytological examination performed?

A short description of SNOT-22, Lund-Mackey and Malm, and their scores, would be useful to make the results understandable.

Finally, it would be useful to indicate the commensal and pathogenic bacteria found in patients with CRS, at least the most abundant genera as described in literature, for example “The Role of Probiotics in Chronic Rhinosinusitis Treatment: An Update of the Current Literature. Bianco MR, Ralli M, Modica DM, Amata M, Poma S, Mattina G, Allegra E. Healthcare (Basel). 2021 Dec 12;9(12):1715.

Author Response

Dear Editor,

We thank you for the opportunity to submit a revised version of the manuscript titled “Expression of Oxidative Stress and Inflammation Related genes in Nasal Mucosa and Nasal Polyps from Patients with Chronic Rhinosinusitis” (ijms-1707010 ) to the International Journal of Molecular Sciences. We are grateful to the reviewers for their comments and valuable suggestions regarding our manuscript. Here, we have enclosed a revised version of the manuscript in response to the reviewer comments, which have been very helpful in improving the manuscript. Included below is a point-by-point description of our responses to reviewer’s comments. Changes made in the manuscript are highlighted in red. We cordially hope that you and the reviewers find this revised manuscript acceptable for publication in your journal.

Martina Mihalj, MD, PhD

Reviewer 2

I thank the editor for invitation to review this manuscript. The study is very interesting, however there are some observations to be made.

  1. The results and discussion are presented in a confusing way. The authors take into consideration numerous parameters of evaluation and continue with numerous and varied comparisons. All this makes the discourse redundant and difficult to follow. Authors should discuss the significance of the data in more detail.

We acknowledge reviewer’s intention to improve our manuscript. In the revised version of the manuscript we propose some correction in the results (lines 131-132, 137-141, 152-154, 164-167, 205-207, 335-336 and 363-366) and discussion (line 417-421, 445-449 and 472-475) sections with the aim of increasing the comprehensibility of the text to the readers.

  1. Section 4.2 Subjects: number of patients shown in the table does not correspond to that in the text. Was a cytological examination performed?

We thank the reviewer for pointing out this discrepancy. This has been changed in the revised version of the manuscript (line 509).

  1. A short description of SNOT-22, Lund-Mackey and Malm, and their scores, would be useful to make the results understandable.

This has now been added in the Material and methods section of the revised version of the manuscript (lines 569-593)

  1. Finally, it would be useful to indicate the commensal and pathogenic bacteria found in patients with CRS, at least the most abundant genera as described in literature, for example “The Role of Probiotics in Chronic Rhinosinusitis Treatment: An Update of the Current Literature. Bianco MR, Ralli M, Modica DM, Amata M, Poma S, Mattina G, Allegra E. Healthcare (Basel). 2021 Dec 12;9(12):1715.

We acknowledge Reviewer’s remark. At the time of study implementation, we have used classical microbiological techniques for isolation and identification of bacteria, based on selection media. At the time, we had no means to identify nasal/sinus microbiome by using MALD TOF or RNASeq technique. Hence, only abundant pathogenic genera and contamination with skin bacteria were identified from sinus swabs of CRSwNP patients, while nasal cavity swabs from control patients remained nominally sterile. Most frequently isolated bacteria in this cohort were Klebsiella pneumonia and Staphylococcus epidermidis, followed by sporadic identification of Staphylococcus Aureus, Escherichia coli, Streptococcus hemolyticus group B, Morganella morganii, Enterobacter spp., Serratia marcescens, Proteus mirabilis and Enterobacter freundii. This has now been included in the revised version of the manuscript (line 118-121).

Reviewer 3 Report

This study appears to be an epoch-making study showing that the target gene expression pattern in the nasal cavity of CRSwNP patients is variously reversed after surgery and postoperative steroid treatment.

It is interesting whether this reversal of the gene expression pattern affected the therapeutic effect after relapse of nasal polyps.

From the data, it can be seen that there is a significant difference in ADCAYP1 etc. between the severe nasal polyp population and the nasal polyp with positive RIST compared to the control. As mentioned by the authors, nasal polyps are allergic (type 2 inflammation) and non-allergic, and the main cells that make up nasal polyps are different, so it is thought that the characteristics of the gene profile are also different accordingly. In that respect, the authors seem to distinguish and verify by the negative or positive of RIST.

As the number of samples increases, new findings may be discovered in the future. The assay method and the experimental method for RNA extraction are described accurately.

Author Response

Dear Editor,

We thank you for the opportunity to submit a revised version of the manuscript titled “Expression of Oxidative Stress and Inflammation Related genes in Nasal Mucosa and Nasal Polyps from Patients with Chronic Rhinosinusitis” (ijms-1707010 ) to the International Journal of Molecular Sciences. We are grateful to the reviewers for their comments and valuable suggestions regarding our manuscript. Here, we have enclosed a revised version of the manuscript in response to the reviewer comments, which have been very helpful in improving the manuscript. Included below is a point-by-point description of our responses to reviewer’s comments. Changes made in the manuscript are highlighted in red. We cordially hope that you and the reviewers find this revised manuscript acceptable for publication in your journal.

Martina Mihalj, MD, PhD

Reviewer 3

This study appears to be an epoch-making study showing that the target gene expression pattern in the nasal cavity of CRSwNP patients is variously reversed after surgery and postoperative steroid treatment.

It is interesting whether this reversal of the gene expression pattern affected the therapeutic effect after relapse of nasal polyps.

From the data, it can be seen that there is a significant difference in ADCAYP1 etc. between the severe nasal polyp population and the nasal polyp with positive RIST compared to the control. As mentioned by the authors, nasal polyps are allergic (type 2 inflammation) and non-allergic, and the main cells that make up nasal polyps are different, so it is thought that the characteristics of the gene profile are also different accordingly. In that respect, the authors seem to distinguish and verify by the negative or positive of RIST.

As the number of samples increases, new findings may be discovered in the future. The assay method and the experimental method for RNA extraction are described accurately.

We thank the reviewer for the kind comments regarding our manuscript.
